# Divergences between Language Models and Human Brains

**Yuchen Zhou    Emmy Liu    Graham Neubig    Michael J. Tarr    Leila Wehbe**
Carnegie Mellon University
{zhouyuchen,emmy,gneubig,michaeltarr,lwehbe}@cmu.edu

## Abstract

Do machines and humans process language in similar ways? Recent research has hinted at the affirmative, showing that human neural activity can be effectively predicted using the internal representations of language models (LMs). Although such results are thought to reflect shared computational principles between LMs and human brains, there are also clear differences in how LMs and humans represent and use language. In this work, we systematically explore the divergences between human and machine language processing by examining the differences between LM representations and human brain responses to language as measured by Magnetoencephalography (MEG) across two datasets in which subjects read and listened to narrative stories. Using an LLM-based data-driven approach, we identify two domains that LMs do not capture well: **social/emotional intelligence** and **physical commonsense**. We validate these findings with human behavioral experiments and hypothesize that the gap is due to insufficient representations of social/emotional and physical knowledge in LMs. Our results show that fine-tuning LMs on these domains can improve their alignment with human brain responses.[1]

## 1  Introduction

Language models (LMs) now demonstrate proficiency that may equal or even surpass human-level performance on tasks including generating text [Brown et al., 2020a], answering questions [Lewis et al., 2019], translating languages [Costa-jussà et al., 2022], and even tasks that necessitate reasoning and inference [Dasgupta et al., 2022]. This has inspired researchers to leverage LM representations to investigate and model the human brain's language system, positing that LMs may serve as reliable proxies for human linguistic processes [Abdou, 2022]. Prior studies have found that human neural activity, as measured by neuroimaging techniques such as fMRI [Jain and Huth, 2018, Toneva and Wehbe, 2019], EEG [Hale et al., 2018], MEG [Wehbe et al., 2014a], and ECoG [Goldstein et al., 2022], can effectively be predicted using representations from language models such as BERT [Devlin et al., 2018] or GPT-2 [Radford et al., 2019]. Robust neural prediction is hypothesized to stem from the shared computational objective of both LMs and the human brain: predicting subsequent words based on prior context [Yamins and DiCarlo, 2016, Schrimpf et al., 2021].

Despite the evident behavioral similarities, the extent to which LMs and human brains align functionally for language processing remains an open question. Essentially, the methods that LMs and humans use to acquire language are very different. LMs learn statistical regularities across massive sets of linguistic symbols, whereas humans rely on applying structured linguistic principles across relatively little input. Additionally, LMs that are confined to linguistic data are likely to fail to ground linguistic symbols in real-world contexts [Harnad, 1990, Bender and Koller, 2020, Bisk et al., 2020a]. Furthermore, the learning environments and goals of LMs and humans are markedly different. While humans communicate through active inquiry, expressing needs, directed communication, and

---

[1]Data and code are available at: `https://github.com/FlamingoZh/divergence_MEG`

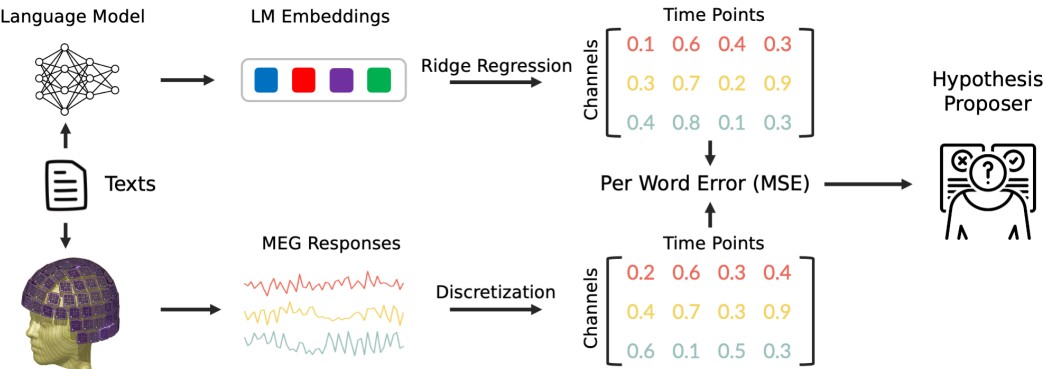

Figure 1: Schematic of our experimental approach. The LM takes as input the current word along with its preceding context to produce the current word's LM embedding. This embedding is then used as input to a ridge regression model to predict the human brain responses associated with the word. The Mean Squared Error (MSE) between the predicted and actual MEG responses is calculated. Finally, an LLM-based hypothesis proposer is employed to formulate natural language hypotheses explaining the divergence between the predicted and actual MEG responses.

scaffolding conversations [Kuhl, 2011], LMs are predominantly trained as passive recipients of raw text data. Consequently, LMs may struggle with comprehending social pragmatics and the nuances of words whose meanings fluctuate across different social contexts [Mahowald et al., 2023].

Previous research exploring the relationship between human and LM language processing has typically focused on the types of linguistic features [Oota et al., 2022a, Sun et al., 2023], neural network architectures [Schrimpf et al., 2021], or training and fine-tuning methods [Sun and Moens, 2023] that may yield better predictions of brain responses. Diverging from this approach, Aw and Toneva [2023] proposed that the divergence between human and LM language processing might stem from LMs' inadequate understanding of texts. They supported this hypothesis by demonstrating that LMs fine-tuned on summarization tasks align more closely with human brain responses. Yet, this hypothesis is only one of many potential explanations. In this work, we adopt a bottom-up, data-driven methodology to systematically investigate the differences between human and machine language processing. Our main contributions are as follows:

1. In contrast to prior studies focusing on the similarities between LMs and human brains, our research emphasizes their differences. We monitor the temporal progression of errors in LM predictions on a word-by-word basis on two datasets with distinct language input modalities (§2).

2. Explaining the prediction errors for every word is challenging due to the vast amount of text. Instead of manually formulating hypotheses, we adopt an LLM-based method that automatically proposes natural language hypotheses to explain the divergent responses between human brains and language models (§3). The top candidate explanations are related to social/emotional intelligence and physical commonsense (§4). We validate these hypotheses with human behavioral experiments.

3. We present evidence that fine-tuning LMs on tasks related to the two identified phenomena can align them more closely with human brain responses. This implies that the observed divergences between LMs and human brains may stem from LMs' inadequate representation of these specific types of knowledge (§5).

## 2 Predictive MEG Model

### 2.1 Data Preparation and Preprocessing

While many studies investigating the correlation between brain responses and language models utilize fMRI recordings (e.g., [Caucheteux et al., 2023, Jain et al., 2020]), the comparatively low temporal resolution of fMRI hinders its ability to accurately capture the processing of individual words. To

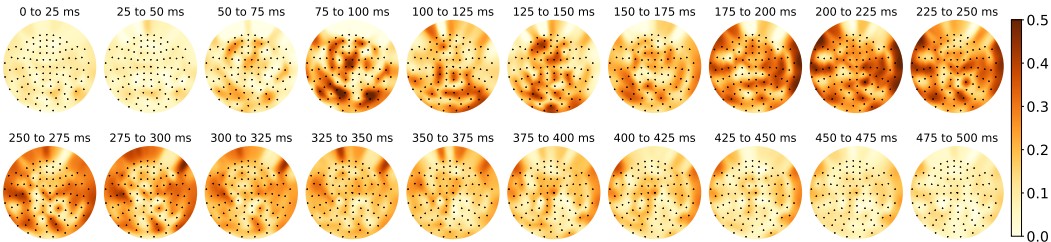

Figure 2: Pearson correlation of actual MEG responses with predicted responses using embeddings from layer 7 of GPT-2 XL on the Harry Potter dataset. The displayed layout is a flattened representation of the helmet-shaped sensor array. Deeper reds indicate more accurate LM predictions. Language regions are well predicted in language processing time windows (refer to §2.4 for more details).

address this limitation, our research employed MEG data. We strategically used two different MEG datasets, each with distinct input modalities, to assess potential variations in the brain's response patterns under these conditions.

The first dataset [Wehbe et al., 2014a] has eight participants reading Chapter 9 of *Harry Potter and the Sorcerer's Stone* (5,176 words) and four participants reading Chapter 10 of the same book (4,475 words) . Each word was exposed on a screen for a fixed duration of 500ms. MEG data were collected on an Elekta NeuroMag MEG with 306 channels at 102 cranial points, and sampled at a rate of 1 kHz. The acquired data underwent preprocessing procedures using the Signal Space Separation (SSS) method [Taulu et al., 2004] and its temporal extension, tSSS [Taulu and Simola, 2006]. The signal was then time-locked with individual words and down-sampled into non-overlapping 25ms time bins. Given the typical low Signal-to-Noise Ratio (SNR) of MEG, we adopted a denoising technique [Ravishankar et al., 2021] that takes advantage of cross-subject correspondences to get an aggregated, denoised version of MEG responses (refer to Appendix A for more details).

To enhance reproducibility and generalizability of our study, we additionally collected MEG data from one participant who listened to six narratives (11,626 words) from The Moth, a platform featuring personal storytelling. These stories were chosen from the stimuli used in a published story listening fMRI dataset [LeBel et al., 2023]. Five of these stories were repeated twice, while one story was repeated five times. The data acquisition was performed using a MEGIN scanner equipped with 306 channels at 102 cranial points. The preprocessing pipeline was similar to that applied to the first dataset. Given that all story repetitions were from the same participant, we averaged the MEG responses for each story's repetitions to enhance SNR without using an alignment method.

## 2.2 Predicting MEG Responses from LM Embeddings

A substantial number of recent studies exploring the correlation between brain responses and LMs have employed GPT-2 [Pasquiou et al., 2022, Caucheteux et al., 2022, 2023, Toneva et al., 2022]. To ensure consistency and comparability with these studies, we utilized the pre-trained GPT-2 XL model with 1.5B parameters, sourced from HuggingFace's *transformers* library [Wolf et al., 2020a], as the backbone language model. Following previous work [Toneva and Wehbe, 2019], for every word $w$, we provided the model with a context consisting of the preceding 19 words. We used the output of hidden layers of the LM, subsequently referred to as LM embeddings, to predict the MEG responses associated with each word (Figure 1). For comparison, we also replicated some analyses on Llama-2 7B [Touvron et al., 2023a] (refer to Appendix D for more details).

Building upon established research that demonstrates the capability of LM embeddings to linearly predict MEG responses [Wehbe et al., 2014a, Jain and Huth, 2018, Caucheteux and King, 2022a], we utilized a linear ridge regression model as the encoding model. Considering the time-correlated nature of MEG data, it was essential to maintain the temporal structure when partitioning the data for training and testing purposes [Yang et al., 2019]. Therefore, we implemented a 10-fold cross-validation procedure that splits the MEG data into 10 continuous chunks. We denote the actual MEG responses as $M$ and LM embeddings as $L$. For split $i$, we set aside one fold as the test set $(M^{i,test}, L^{i,test})$ and fitted a ridge regression model with weight matrix $W^i$ and bias $b^i$ using the remaining folds, denoted as $(M^{i,train}, L^{i,train})$. The regularization parameters were chosen via

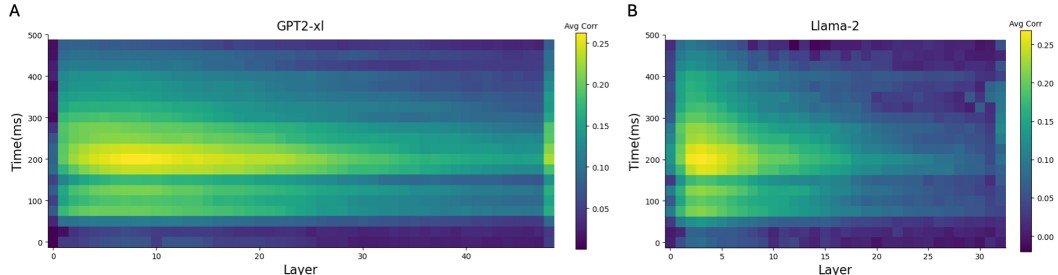

Figure 3: Pearson correlation between actual MEG responses and predicted responses from (A) GPT-2 XL and (B) Llama-2 across LM layers and time after word onset on the Harry Potter dataset. Both models exhibit high correlations in early and intermediate layers at around 200ms. Correlation is computed across words and averaged across MEG channels.

nested cross-validation. Following model training, we applied the trained weight matrix and bias to predict the brain responses from the LM outputs for the test set:

$$\hat{M}^{i,test} = L^{i,test}\hat{W}^i + \hat{b}^i$$

Finally, the test predictions from all folds were concatenated to form the comprehensive prediction of MEG responses from the LM:

$$\hat{M} = concat[\hat{M}^{i,test}]$$

This process is performed for each of the different time windows relative to word onset.

## 2.3 Best Language Model Layer for Predicting MEG Responses

Prior research has shown that intermediate layers of language models often best predict human brain responses [Toneva and Wehbe, 2019, Jain and Huth, 2018, Oota et al., 2022b]. Therefore, we selected the layer that best predicts brain responses. Figure 3 illustrates the Pearson correlation between actual MEG responses and those predicted by LM embeddings across layers and time points relative to word onset. We used the average correlation across all words and time windows as the metric to select the best layer. Echoing previous findings, we confirmed that intermediate layers exhibit higher correlations, with layer 7 being the best at predicting brain responses in GPT-2 XL. Similarly, for Llama-2, layer 3 was identified as the most predictive.

## 2.4 Spatio-temporal Pattern of Predictions

As a sanity check, we examined if the predictive model can effectively predict the brain areas and time course of language processing. These areas include the inferior frontal gyrus, superior temporal gyrus, certain sections of the middle temporal gyrus, and angular gyrus [Blank et al., 2016, Rogalsky et al., 2015, Sahin et al., 2009, Brennan and Pylkkänen, 2012, Friederici, 2002, Visser et al., 2010, Rogalsky and Hickok, 2009].

As shown in Figure 2, we observe a temporal progression of accurately predicted areas after word onset. The prediction performance peaks first in the occipital lobe between 75-100ms. Given that LM embeddings encode information (e.g., word frequency) correlated to the number of letters in a word and MEG is sensitive to abrupt changes in visual inputs, we attribute this early peak to the initial visual perception of a word. This is followed by heightened prediction performance in the bilateral temporal lobe between 175-250ms, when we expect semantic processing to start. This observation aligns with previous research indicating that most language experiments with naturalistic stimuli reveal bilateral language representations [Wehbe et al., 2014b, Huth et al., 2016, Deniz et al., 2019, Toneva et al., 2022]. Finally, between 250-375ms, the anterior temporal lobe and frontal lobe show increased prediction performance, which is likely related to further semantic processing. This sequential pattern of prediction performance replicates the spatio-temporal dynamics of language processing found in previous literature [Wehbe et al., 2014a, Toneva et al., 2022].

Table 1: Top 10 hypotheses generated from the best layer of GPT-2 XL on the Harry Potter dataset

| Hypothesis | Validity | p-value |
|---|---|---|
| have a high level of emotional intensity | 0.250 | 0.010 |
| involve complex sentence structures or grammar | 0.250 | 0.015 |
| include emotional language or descriptions | 0.238 | 0.008 |
| have a high level of tension or conflict | 0.237 | 0.023 |
| have characters using body language or non-verbal cues | 0.225 | 0.032 |
| are emotionally charged, making it challenging for language models to accurately interpret the intended tone or sentiment | 0.213 | 0.020 |
| include conflicts between characters | 0.200 | 0.035 |
| have characters interacting with their environment | 0.188 | 0.059 |
| have complex sentence structures | 0.175 | 0.081 |
| have dialogue between characters with varying emotions | 0.175 | 0.022 |

Table 2: Top 10 hypotheses generated from the best layer of GPT-2 XL on the Moth dataset

| Hypothesis | Validity | p-value |
|---|---|---|
| contain elements of fiction or exaggeration | 0.212 | 0.012 |
| feature emotional or dramatic language | 0.150 | 0.090 |
| refer to cultural or societal norms | 0.138 | 0.107 |
| include sensory details or imagery | 0.137 | 0.107 |
| have strong emotional or dramatic content | 0.100 | 0.173 |
| show a lack of coherence or logical flow | 0.100 | 0.111 |
| contain elements of surprise and unpredictability | 0.094 | 0.201 |
| contain emotional, personal narratives | 0.088 | 0.201 |
| use idiomatic expressions or figurative language | 0.088 | 0.178 |
| refer to specific events or incidents | 0.087 | 0.237 |

## 3 Identifying Phenomena of Interest

Our objective is to investigate the elements of MEG responses that cannot be well explained by the LM. We work with an average of cleaned MEG responses from a group of subjects and multiple trials, which illustrate the common elements of language processing across individuals. Therefore, for words where MEG responses are not well predicted, it is likely that this marks a genuine divergence between human brains and the LM. It is important to clarify that our approach trains an encoding model to predict human brain responses based on language model outputs, rather than the reverse. This means our methodology identifies information that is captured by MEG but is not present in the language model, rather than information captured by the language model but is not present in MEG responses.

Leveraging the high temporal resolution of MEG, we computed the Mean Squared Errors (MSEs) between actual and predicted MEG responses for each individual word on channels that demonstrated statistically significant correlations (one-sided, $p$=0.001). For word $w$,

$$MSE(w) = \frac{1}{|S|} \cdot \sum_{i \in S} (\hat{M}(w)_i - M(w)_i)^2 \tag{1}$$

where $S$ is the set of significant channels.

### 3.1 Automatically Discovering Differences between Brain Responses and LM Predictions

Given the vast amount of text, manual pattern discovery becomes challenging. Figure 7 presents sample sentences color-coded based on prediction error, illustrating the challenges in formulating hypotheses from observations.

To discover subtle differences between MEG responses and LM predictions, we used a method that automatically describes differences between text corpora using proposer and verifier LMs [Zhong et al., 2023]. This system consists of first prompting an LLM (GPT-3; Brown et al. [2020b]) with a

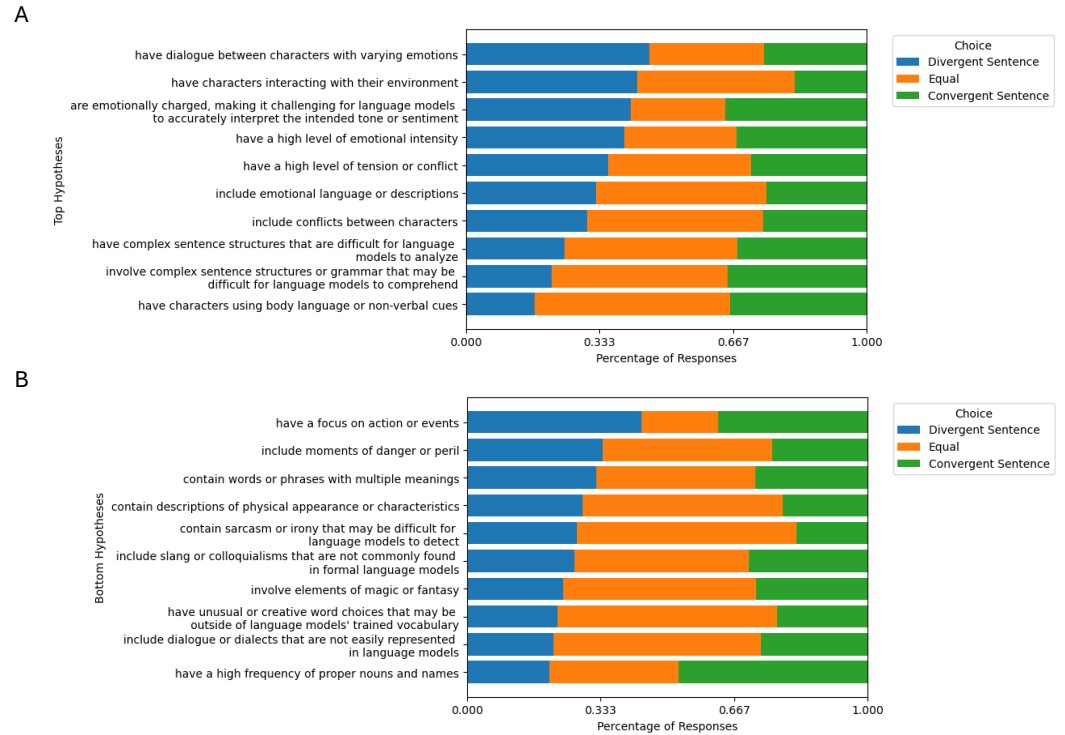

Figure 4: Distribution of human responses for (A) the top 10 and (B) the bottom 10 hypotheses, ranked by the percentage of 'Divergent Sentence' responses.

number of samples from two corpora ($D_0, D_1$) to generate many hypotheses on how the first corpus differs from the second, and then using a fine-tuned validator model (FLAN-T5-XXL; Chung et al. [2022]) to validate how often each proposed hypothesis is true based on pairs from each corpus sampled from a held-out set. Specifically, the verifier is presented with a prompt containing two sentences from $D_0$ and $D_1$, and asked whether or not the hypothesis is true, and this is repeated across the development set for each hypothesis. For the exact prompts used in the proposer and verifier, please refer to Appendix E. Sentences were then ranked based on their mean MSE. The top 100 least-predicted sentences constituted the $D0$ set, while the 100 well-predicted sentences constituted the $D1$ set. This process of hypothesis proposal and verification was repeated across 3 cross-validation folds.

## 3.2 Proposed Hypotheses

The top ten hypotheses from the Harry Potter dataset ranked by validity[2] are listed in Table 1. We identified two primary differences between the language model and the human brain: firstly, the processing of social and emotional information, and secondly, the capacity for interaction with the surrounding environment. These are henceforth referred to as **social/emotional intelligence** and **physical commonsense**, respectively. Importantly, these hypotheses resonate with conclusions drawn in prior research, as detailed in §4. Similarly, we ran the hypothesis proposer on the Moth dataset. This replication produced slightly varied but fundamentally similar topics to those discovered in the Harry Potter dataset (Table 2). This congruence across datasets with different input modalities aligns with previous research showing that after initial sensory processing, the brain's language processing is consistent across reading and listening [Deniz et al., 2019, Chen et al., 2023].

---

[2]Validity measures the difference in certainty that the hypothesis is true between the two corpora, see Zhong et al. [2023] for more details.

### 3.3 Manual Hypothesis Verification

We conducted an experiment involving human participants for additional validation of our hypotheses. We gathered data from 10 participants using Qualtrics, resulting in a collection of 1,400 trials. In each trial, participants were presented with a hypothesis selected either from the top 10 or bottom 10 hypotheses generated from the Harry Potter dataset, along with a pair of sentences — one from $D0$ and the other from $D1$ — in a randomized order. The task for participants was to determine which sentence aligned more closely with the given hypothesis, choosing between "More True for Sentence A", "More True for Sentence B", or "Equally true".

The response distribution for each hypothesis is shown in Figure 4. Note that a given hypothesis is not expected to apply to all divergent sentences (e.g., it might suffice for a sentence to be emotionally intense or grammatically complex to be divergent). If a hypothesis does not align with a sentence from the divergent set, participants should show no preference between the two sentences presented. Chi-square analysis revealed statistically significant differences in the distribution of responses between the top and bottom hypotheses ($p = 0.024$). A preference towards divergent sentences was observed in the top hypotheses condition while a preference towards "Equally True" was observed in the bottom hypothesis condition. This pattern can be attributed to the role of the proposer, which was instructed to generate hypotheses that effectively distinguish $D_0$ (comprising divergent sentences) from $D_1$. As a result, the bottom hypotheses tend to be those that fail to differentiate between $D_0$ and $D_1$, rather than those that are more explanatory of $D_1$ compared to $D_0$ (refer to Appendix F for more details).

## 4 Selected Phenomena

Comprehending social/emotional and physical commonsense requires humans use a broad spectrum of contextual knowledge. We briefly discuss the insights and challenges highlighted in the existing neuropsychological and NLP literature regarding these domains.

**Human social and emotional intelligence** requires both introspection and predicting the feelings of others [Salovey and Mayer, 1990]. Neuropsychological research on social cognition has identified a network of brain regions that support understanding other people's intentions, actions, and emotions [Saxe et al., 2006]. Crucially, emotions are intrinsic to the human experience and pervasively interact with other mental facilities, including language [Satpute and Lindquist, 2021]. As such, creating agents with social and emotional intelligence has been a longstanding goal of NLP [Gunning, 2018, Paiva et al., 2021]. However, LLMs still fall short of human abilities for inferring the mental states and emotions of others ("theory-of-mind" tasks) [Sap et al., 2022].

**Physical commonsense** refers to knowledge about the physical properties of everyday objects and physical phenomena [Forbes et al., 2019, Bisk et al., 2020b]. From a neuropsychological perspective, language is not the primary means through which humans acquire commonsense physical knowledge. Instead, humans rely on sensory inputs and interactions with their environment [Baillargeon, 1994]. Notably, the category of a physical object affects which brain regions are recruited when interacting with that object. For example, interacting with people activates the theory of mind areas [Saxe et al., 2006], the visual face areas [Sergent et al., 1992, Kanwisher et al., 1997], and body areas [Downing et al., 2001], interacting with corridors while navigating recruits the visual place [Epstein and Kanwisher, 1998] and spatial navigation areas, and interacting with tools recruits the dorsal object-processing stream [Almeida et al., 2010]. Interestingly, reading about these domains has also been found to recruit these same visual regions [Wehbe et al., 2014b, Huth et al., 2016]. Given how physical commonsense knowledge is acquired, it is not surprising that, within NLP, this domain poses a challenge for language models. While these models can potentially learn representations capturing specific physical properties of the world, such as an object's color or a game board's state [Abdou et al., 2021, Li et al., 2023], it remains unclear whether purely text-based representations can capture the richness and complexity of physical commonsense as exhibited by humans [Forbes et al., 2019, Bisk et al., 2020b].

## 5 Improving Brain Alignment via Fine-tuning

We hypothesize that the inability of language models to accurately predict associated brain responses stems from their inadequate representations of social/emotional understanding and physical world

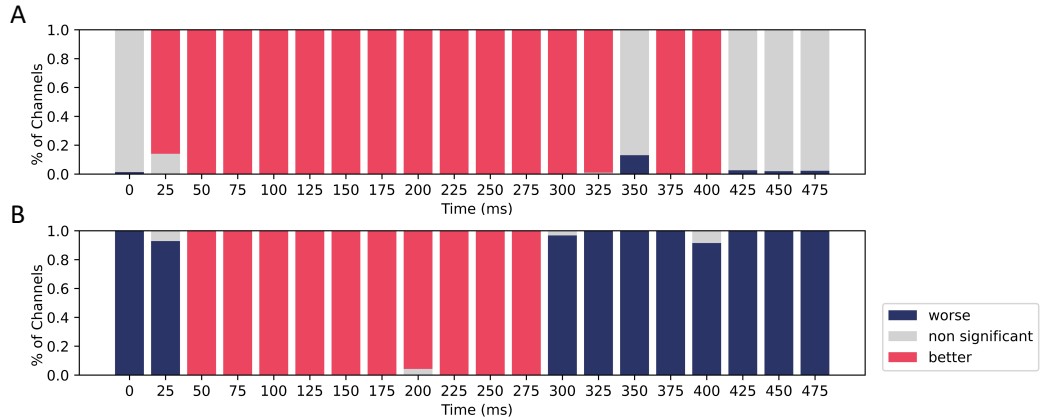

Figure 5: Performance comparison of the base model with models fine-tuned on (A) social and (B) physical datasets. Each panel's y-axis shows the percentage of channels in the fine-tuned model with better, worse, or non-significantly different performance (measured by Pearson correlation) compared to the base model. Fine-tuned models outperform the base model during language processing time windows. Refer to Appendix K for a detailed view of each MEG channel plotted.

knowledge. Drawing inspiration from Aw and Toneva [2023], we fine-tuned the GPT-2 XL model on datasets specific to the two phenomena to examine if targeted fine-tuning could enhance the model's alignment with brain activity.

Furthermore, we examined whether domain-specific fine-tuning would specifically bolster the model's capability in predicting MEG responses associated with words from that domain, as compared to words outside that domain. To this end, we recruited three raters to annotate Chapter 9 of *Harry Potter* across the two domains. We release these annotations as a resource for the dataset to facilitate further analysis. Details on the annotation process can be found in Appendix H. Examples of each phenomenon within the *Harry Potter* text can be found in Appendix I.

## 5.1 Datasets

**Social/Emotional Intelligence** We study social and emotional intelligence using the Social IQa dataset [Sap et al., 2019]. This dataset contains questions about people's feelings and their social implications.

**Physical Commonsense** We study physical commonsense using the PiQA dataset [Bisk et al., 2020b]. This dataset contains goal-driven questions based on everyday situations. These questions were taken from the website instructables.com, where people share DIY project instructions.

We also provide examples from each dataset in Table 3.

Table 3: Datasets for Fine-Tuning with Sample Questions and Answers (Correct Answer in Bold)

| Dataset | Type | Num train | Options | Sample question | Sample answers |
|---------|------|-----------|---------|-----------------|----------------|
| Social IQa | Social/Emotion | 33.4k | 3 | Sydney had so much pent up emotion, they burst into tears at work. How would Sydney feel afterwards? | 1. affected
2. **like they released their tension**
3. worse |
| PiQA | Physical | 16.1k | 2 | When boiling butter, when it's ready, you can | 1. Pour it onto a plate
2. **Pour it into a jar** |

## 5.2 Fine-tuning Setup

In order to keep the architecture of fine-tuned models consistent with the base model, we format the multiple choice task as $N$ language modeling tasks, where $N$ is the number of options. Specifically, for the combined context and question $x$, we directly concatenate each possible multiple-choice answer $\{y_1, ..., y_N\}$ to $x$ to form $N$ different sentences. After passing the concatenated sequences through the model, we sum the logits of all tokens corresponding to each multiple-choice option to obtain a score proportional to its log-likelihood. These scores are then gathered into a size $(1, N)$ tensor, and cross-entropy loss relative to the correct multiple choice answer is used to train the model. Further details on the fine-tuning setup can be found in Appendix G.

## 5.3 Comparing Fine-tuned Models with the Base Model

We evaluated the fine-tuned GPT-2 XL model on the Harry Potter dataset. To identify channels with statistically significant differences between the base and fine-tuned model, we calculated empirical $p$-values by comparing the true correlation value with 10,000 simulated ones obtained by permuting the brain data. Details of the algorithm can be found in Appendix J.

**Fine-tuned models are better aligned with the brain on both tasks.** As illustrated in Figure 5A, the model fine-tuned on the social dataset exceeds the base model in performance across the majority of channels within the 50ms to 300ms time interval post word onset. Notably, this interval corresponds to the language processing time windows, as identified in §2.4. In a similar vein, the fine-tuned physical model exceeds the base model's performance in almost all channels during the 50-275ms interval post word onset (Figure 5B). However, interestingly, almost all channels are worse than the base model outside this time window. This time selectivity may indicate that the improvements of the fine-tuned model are tailored towards linguistic comprehension rather than broader brain functionalities.

**Fine-tuning improves alignment more for words annotated with that category.** We compared the reduction in prediction error for words annotated within each category and words outside each category by computing the difference in MSE between the model fine-tuned on the corresponding task and the base model. As demonstrated in Figure 6A, prediction errors for social words exhibit a significant reduction compared to non-social words 200-275ms post word onset. Additionally, there is a significant improvement in MSE for physical words over non-physical words 150-225ms post word onset (Figure 6B). We also ran additional control experiments to check if MSE improvement is specific to words that match the category of the dataset on which the model was fine-tuned. Specifically, we evaluated the prediction improvement of physical words on the model fine-tuned on the social dataset, and vice versa (Appendix L).

**Improvements are not related to increased language-modeling ability.** Prior work has found that LMs with lower perplexity can better predict brain activity [Schrimpf et al., 2021]. Therefore, additional fine-tuning may have improved the language model's ability to perform the LM task in general, leading to improved alignment. To rule out this possibility, we performed 3-fold cross-validation on *Harry Potter and the Sorcerer's Stone*, excluding Chapters 9 and 10, which were used as data in this study. We trained the base model, as well as the fine-tuned models, on the train set in each fold with the language modeling objective, and found that the final average losses on the test sets were similar (See Appendix M for details).

## 6 Related Work

Numerous studies have found that LM hidden states can linearly map onto human brain responses to speech and text measured by MEG, EEG, and fMRI [Wehbe et al., 2014a, Hale et al., 2018, Jain and Huth, 2018, Abnar et al., 2019, Jat et al., 2019, Gauthier and Levy, 2019, Toneva and Wehbe, 2019, Caucheteux and King, 2022a, Jain et al., 2020, Toneva et al., 2022, Aw and Toneva, 2022].

At a more foundational level, studies have identified shared computational principles between LMs and human brains. Evidence suggests that both human brains and LMs are perpetually engaged in predicting the subsequent word [Schrimpf et al., 2021]. LM surprisal is found to be positively correlated with brain activation, reaching its peak approximately 400 ms post word onset [Goldstein et al., 2022]. This aligns well with N400, which denotes a decline in brain activation upon encountering unexpected words around 400 ms after word onset [Lau et al., 2009, Parviz et al., 2011, Halgren et al.,

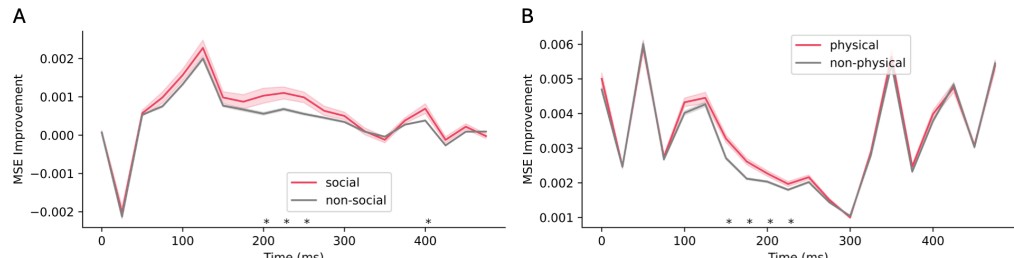

Figure 6: Comparison of improved MSE between (A) social and (B) physical words and those outside each category evaluated on models fine-tuned on corresponding datasets. Positive values denote lower MSEs in the fine-tuned model. Shaded region indicates standard error. Asterisks denote time points with significant differences between the two groups (Student's t-test with FDR correction, $p$=0.05).

2002]. Moreover, LM representations can predict the hierarchy of brain responses [Caucheteux and King, 2022b, Caucheteux et al., 2023]. Despite this, Antonello and Huth [2022] have pointed out that a high correlation between brain activity and LMs does not necessarily imply that they operate under similar computational principles.

We not only observe this LM-brain alignment but can also actively intervene in it. Research has demonstrated that the alignment between LMs and human brains can be improved by task-specific fine-tuning. A notable instance is the study by Schwartz et al. [2019], where the fine-tuning of BERT using both fMRI and MEG signals enhanced its ability to predict fMRI responses. Importantly, this improvement was not participant-specific and could be transferred to hold-out individuals. Another study [Aw and Toneva, 2023] showed that task-oriented fine-tuning, particularly for narrative summarization, also facilitated better alignment with brain activity. Furthermore, altering the architecture of BERT such that it aligns better with the brain improves its performance on downstream NLP tasks [Toneva and Wehbe, 2019]. These findings suggest a potentially symbiotic relationship between enhancing task performance in LMs and boosting their alignment with brain responses.

## 7 Conclusions, Limitations, and Future Work

We explore a critical question connecting language models with human neural activity: How do LMs differ from human brains in processing language? We employed an LLM-based approach to automatically propose hypotheses explaining the elements of human brain responses that cannot be well explained by language models. Social/emotional intelligence and physical commonsense emerged as the two dominant themes. After fine-tuning GPT-2 XL on datasets related to these themes, we observed an improved alignment between LM predictions and human brain responses.

Limited by the availability of datasets with aligned brain data, our study was conducted on a relatively narrow range of texts. While we observed consistent patterns across two language modalities, it is important to note that both datasets utilized were exclusively narrative stories. This limited scope raises the possibility that additional, undetected divergences exist, potentially obscured by the quantity of text and the methodology employed for hypothesis generation from the sentences. By developing more robust tools for pattern discovery and incorporating a wider array of textual materials, our approach can be adapted to more comprehensively address the question in future studies.

## Acknowledgments and Disclosure of Funding

This work is supported by the James S. McDonnell Foundation. We also acknowledge the support of the National Sciences and Engineering Research Council of Canada (NSERC) through a post-graduate fellowship given to EL (PGSD). This work was also partially supported by the National Institute On Deafness And Other Communication Disorders of the National Institutes of Health under Award Number R01DC020088. The content is solely the responsibility of the authors and does not necessarily represent the official views of the National Institutes of Health.

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

# A  MEG Denoising

Because of the typical low Signal-to-Noise Ratio (SNR) associated with MEG, we adopted a denoising technique [Ravishankar et al., 2021] that takes advantage of cross-subject correspondences to get an aggregated, denoised version of MEG responses. Specifically, this process involves modeling the MEG responses $M_t$ of subject $t$ as a linear function of the MEG responses $M_s$ from a source subject $s$:

$$\hat{M}_{t \leftarrow s} = \hat{W}_{t \leftarrow s} M_s + \hat{b}_{t \leftarrow s}$$

We estimated the target subject's MEG responses from all other subjects:

$$\hat{M}_t = \frac{1}{N-1} \sum_{s \in S, s \neq t} \hat{M}_{t \leftarrow s}$$

where $S$ is the set of subjects and $N$ is the number of subjects. These individual estimates are then aggregated to generate a denoised version of MEG responses:

$$\hat{M} = \frac{1}{N} \sum_{s \in S} \hat{M}_t$$

# B  Sample Sentences

Given the vast amount of text in the datasets, manually discovering patterns becomes challenging. Figure 7 provides an illustrative example by presenting a set of sample sentences that are color-coded based on the magnitude of their prediction error. This visualization demonstrates the complexity involved in formulating hypotheses from observations.

1. He had been looking forward to learning to fly more than anything else.

2. "Of course he has," said Ron, wheeling around.

3. But Neville, nervous and jumpy and frightened of being left on the ground, pushed off hard before the whistle had touched Madam Hooch's lips.

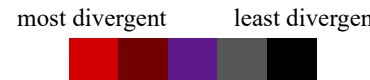

Figure 7: Sample sentences from the Harry Potter dataset, with colors indicating prediction error levels. Each of the five colors corresponds to a 20-percentile range of words from the entire dataset.

# C  Additional Results on GPT-2 XL

## C.1  Results on Last Layer

In addition to the best layer, we also performed analyses on the last layer of the language model.

### C.1.1  Spatial-Temporal Patterns of Predictions

The spatial-temporal pattern of predictions observed in the last layer (Figure 8) is similar to that of the best layer, However, there is a notable difference in the magnitude of the values. Specifically, the maximum correlation in the last layer is lower, decreasing from $0.53$ in the best layer to $0.43$.

### C.1.2  Proposed Hypotheses

We also generate hypotheses from the predictions of the last layer (Table 4). These hypotheses exhibit similarities with those derived from the best-performing layer, notably in their inclusion of emotions

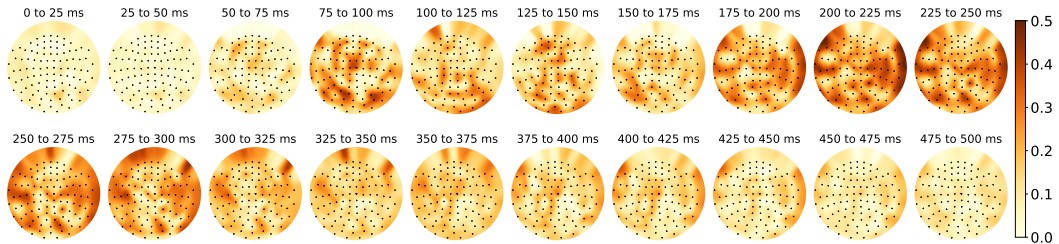

Figure 8: Pearson correlation of actual MEG responses with those predicted by LM embedding from the last layer of GPT-2 XL (evaluated on the test set). The displayed layout is a flattened representation of the helmet-shaped sensor array. Deeper reds indicate more accurate LM predictions. Language regions are effectively predicted in language processing time windows (refer to §2.4 for more details).

and social interactions. However, a distinctive aspect of these hypotheses is their association with supernatural and magical elements. Additionally, we observe the emergence of figurative language, aligning with previous research that indicates language models underperform humans in both the interpretation and generation of figurative language [Chakrabarty et al., 2022, Liu et al., 2022] and the correct representation of idiomatic phrases [Dankers et al., Liu and Neubig, 2022].

Table 4: Top 10 hypotheses generated from the last layer of GPT-2 XL on the Harry Potter dataset

| Hypothesis | Validity | *p*-value |
| --- | --- | --- |
| contain descriptions of unusual settings or creatures | 0.1750 | 0.0754 |
| has a lot of dialogue, with characters speaking to each other | 0.1719 | 0.0855 |
| contain figurative language | 0.1367 | 0.1409 |
| contain rhetorical questions or exclamations | 0.1125 | 0.1685 |
| contains references to obscure facts or trivia, such as the longest game of Quidditch | 0.1094 | 0.0627 |
| mentions the unknown or unexpected, such as an unknown creature or a surprise announcement | 0.1094 | 0.1856 |
| contain references to the emotions of characters | 0.1062 | 0.1996 |
| contain references to the supernatural | 0.0875 | 0.1827 |
| mentions dangerous creatures and events, such as trolls and duels | 0.0813 | 0.2383 |
| contain references to magic | 0.0688 | 0.2550 |

# D   Replication on Llama-2 7B

Although GPT-2 is a widely used language model in brain research, it's not the latest model in the field. Models with more parameters and advanced training methods could show different results. Therefore, we replicated some analyses on Llama-2 [Touvron et al., 2023b]. We used the implementation in the HuggingFace library [Wolf et al., 2020b] with 7B parameters. As Figure 3B shows, early layers exhibit high correlations, with layer 3 identified as having the highest correlation.

## D.1   Spatio-Temporal Patterns of Predictions

The spatial-temporal pattern of predictions in layer 3 of Llama-2 (Figure 9) closely resembles those found in GPT-2 XL. This similarity implies that both language models effectively capture the representations of words.

## D.2   Proposed Hypotheses

Hypotheses from the predictions of layer 3 of Llama-2 7B can be found in Table 5. Interestingly, the focus of these hypotheses is primarily on physical objects and events. In comparison to the hypotheses produced by GPT-2 XL, there is a notable absence of social and emotional aspects, suggesting that Llama-2 7B could have a more advanced comprehension of social and emotional contexts.

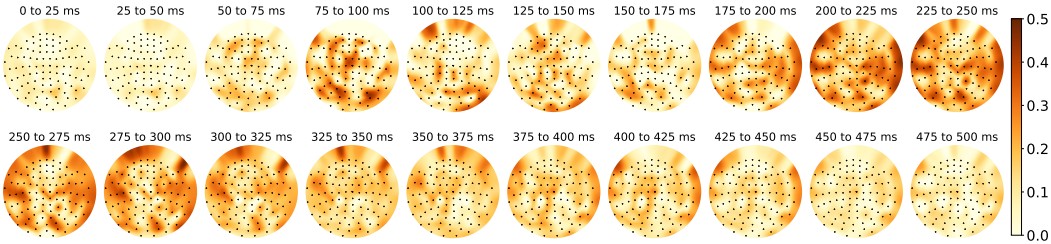

Figure 9: Pearson correlation of actual MEG responses with those predicted by LM embedding from the best layer (layer 3) of Llama-2 (evaluated on the test set). The displayed layout is a flattened representation of the helmet-shaped sensor array. Deeper reds indicate more accurate LM predictions. Language regions are effectively predicted in language processing time windows (refer to §2.4 for more details).

Table 5: Top 10 hypotheses generated from the best layer of Llama-2 on the Harry Potter dataset

| Hypothesis | Validity | *p*-value |
|---|---|---|
| involve action or movement, such as running or tiptoeing | 0.300 | 0.005 |
| refer to specific events or actions, such as a flying lesson or a spell not working | 0.237 | 0.029 |
| refer to specific objects or locations, such as the front steps or the trophy room | 0.237 | 0.013 |
| describe physical actions or movements | 0.175 | 0.081 |
| discuss or describe dangerous or frightening situations | 0.150 | 0.056 |
| include actions or physical movements | 0.150 | 0.117 |
| contain words or phrases that are specific to the wizarding world | 0.140 | 0.130 |
| have a sense of chaos or disorder | 0.137 | 0.040 |
| have a high level of tension or suspense | 0.137 | 0.128 |
| include words or phrases that are specific to the wizarding world | 0.127 | 0.152 |

# E    Proposer and Verifier Prompts

The prompt for the proposer is:

> {A_block}
> {B_block}
> The dataset includes two chapters from "Harry Potter and the Sorcerer's Stone". The two groups are generated based on the difference between language model and human responses to these sentences. The Group A snippets sentences where language models and humans show divergent responses, while the Group B snippets sentences where language models and humans show similar responses.
>
> I am a literary analyst investigating the characteristics of words. My goal is to figure out which sentences induce different responses for language models and human responses.
>
> Please write a list of hypotheses about the datapoints from Group A (listed by bullet points "-"). Each hypothesis should be formatted as a sentence fragment. Here are three examples.
> - "{example_hypothesis_1}"
> - "{example_hypothesis_2}"
> - "{example_hypothesis_3}"
> Based on the two sentence groups (A and B) from the above, more sentences in Group A ...

The prompt for the validator is:

> Check whether the TEXT satisfies a PROPERTY. Respond with Yes or No. When uncertain, output No.
> Now complete the following example -
> input: PROPERTY: {hypothesis}

TEXT: {text}
output:

# F   Manual Hypothesis Verification

## F.1   Experiment Setup

We recruited 10 participants through Qualtrics. Of these, 9 participants completed 100 trials each, while one participant completed 500 trials. In each trial, participants were presented with a hypothesis selected either from the top 10 or bottom 10 hypotheses generated from the Harry Potter dataset, along with a pair of sentences — one from $D0$ (the divergent sentence set) and the other from $D1$ (the convergent sentence set) — in a randomized order. The task for participants was to determine which sentence aligned more closely with the given hypothesis, choosing between "More True for Sentence A", "More True for Sentence B", or "Equally true". Note that a given hypothesis is not expected to apply to all divergent sentences (e.g., it might suffice for a sentence to be emotionally intense or grammatically complex to be divergent) so it is expected that some of the responses will be "Equally true". A screenshot of the experiment can be found in Figure 10.

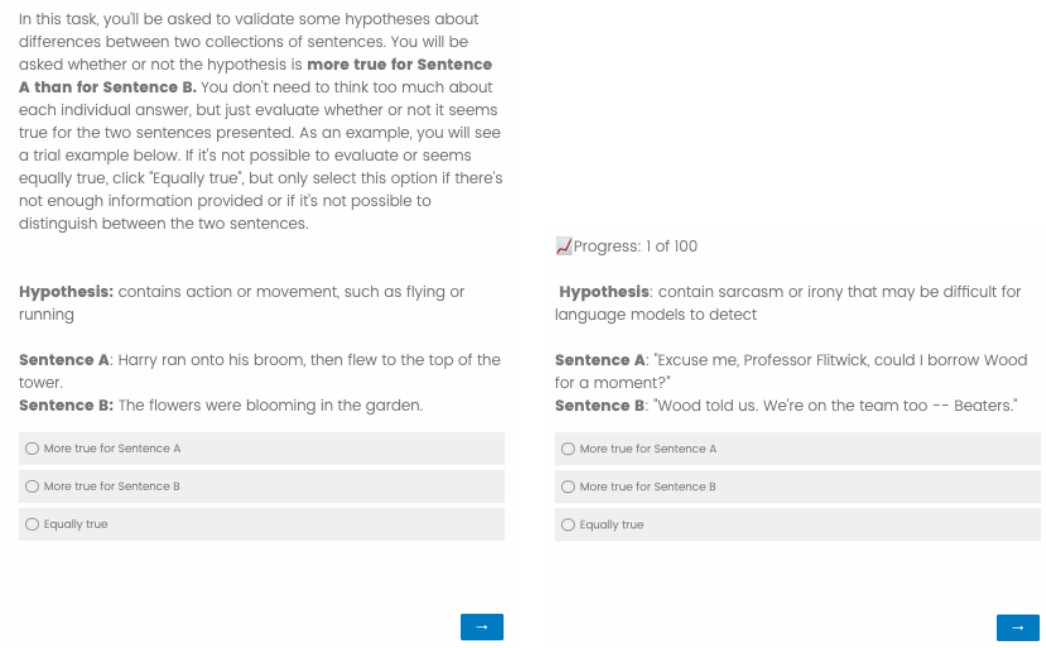

Figure 10: Screenshots of the experiment.

## F.2   Results

We constructed a contingency table with dimensions (Top Hypothesis, Bottom Hypothesis) by (Prefer Divergent, Equal, Prefer Convergent) (Table 6). A binomial test conducted on the contingency table (looking only at the "Prefer Divergent" and "Prefer Convergent" responses) showed that divergent sentences were more likely to be chosen over convergent sentences on average ($p < 10^{-10}$). A Chi-square test revealed statistically significant differences in the distribution of responses between the top and bottom hypotheses ($p = 0.024$). Additionally, we utilized the Chi-square test to compare the frequency of "Prefer Divergent," "Equal," and "Prefer Convergent" responses in two conditions. Notably, a preference towards divergent sentences was observed in the top hypotheses condition compared to the bottom hypotheses ($p = 0.093$). In contrast, in the bottom hypothesis condition, there was a marked preference for "equally true" ($p = 0.008$). No significant difference was observed in the preference for convergent sentences between the conditions ($p = 0.676$).

Table 6: Contingency Table for Human Responses

|  | Divergent | Equal | Convergent |
|---|---|---|---|
| **Top** | 377 | 122 | 209 |
| **Bottom** | 336 | 159 | 196 |

# G Fine-tuning details

## G.1 Computational Details

GPT-2 XL was trained separately on each of the two datasets in subsection 5.1 on 4 A6000 GPUs with 16-bit quantization and a batch size of 1 per GPU. Deepspeed with ZeRo stage 2 optimization was used in order to parallelize training [Rasley et al., 2020]. The Adam optimizer was used with a learning rate of 1e-5, betas of $(0.9, 0.999)$, epsilon of 1e-8, and no weight decay. Models were trained with early stopping with a patience of 3 [Kingma and Ba, 2017].

## G.2 Multiple-choice training

Let $x_i$ represent the concatenation of the context, if applicable, and the question. Then for each answer choice $y_i$, we concatenate it with the question and context, and feed it to the model to obtain a sequence of logits.

$$\ell_i = \text{Model}(x_i \oplus y_i) \tag{2}$$

Then we sum the logits corresponding to the sequence, where $t \in [1, T]$ represents the total length of $x_i \oplus y_i$.

$$\text{score}_i = \sum_{t=1}^{T} \ell_{i,t} \tag{3}$$

Finally, we take the cross-entropy loss of these values relative to a one-hot encoding of the correct option, where $t_i = 1$ if option $i$ is correct, or else 0.

$$P_i = \frac{\exp(\text{logit}_i)}{\sum_{j=1}^{N} \exp(\text{logit}_j)}$$

$$L = -\sum_{i=1}^{N} t_i \log(P_i)$$

### G.2.1 Performance on Multiple-Choice Datasets

We note that the performance of the final model may not approach that of GPT-2 XL fine-tuned with an output size of $N$ denoting each option, as we keep the output dimension the same as the size of the vocabulary. However, we report the final accuracy achieved by each model on the original datasets here.

Table 7: Summary of model performance

| Dataset | Best epoch | Accuracy (%) | Baseline (random) accuracy |
|---|---|---|---|
| Social IQa | 4 | 54.86% | 33.33% |
| PiQA | 1 | 73.88% | 50.00% |

# H  Annotations

To decide which category a word belongs to, we employed three raters who used binary coding to indicate if a word belonged to the target category. The consistency among raters was evaluated using Krippendorff's alpha. Their consistency was 0.54 for social/emotion and 0.87 for physical. Finally, if at least two out of the three people annotated a word as fitting a category, we counted it as belonging to that category.

## H.1  Annotation Guidelines

### H.1.1  Social/Emotional Intelligence

- Include words that depict the emotions of characters and/or social interactions.

- Exclude words that suggest emotions or social interactions indirectly. For instance, "slam the door" shouldn't be annotated.

### H.1.2  Physical commonsense

- Annotate words referring to tangible entities, such as characters (people) and physical objects.

- Do not annotate words that represent concrete ideas but lack physical substance, like "laughter".

- Pronouns should also be excluded.

# I  Examples of phenomena in Harry Potter

We give some examples of the two phenomena in the dataset according to the annotations. Words of that category are marked in bold.

## I.1  Social/Emotional

- Harry had never believed he would meet a boy he **hated** more than Dudley.

- Hermione Granger was almost as **nervous** about flying as Neville was.

- But Neville, **nervous** and **jumpy** and **frightened** of being left on the ground, pushed off hard before the whistle had touched Madam Hooch's lips.

## I.2  Physical Commonsense

- Up the **front steps**, up the **marble staircase** inside, and still **Professor McGonagall** didn't say a word to him.

- **Ron** had a piece of **steak** and **kidney pie** halfway to his **mouth**, but he'd forgotten all about it.

- They pulled on their **bathrobes**, picked up their **wands**, and crept across the **tower room**, down the **spiral staircase**, and into the **Gryffindor common room**.

# J  Algorithm for Permutation Test

To identify channels on which the performance of the fine-tuned model and the base model has statistically significant differences, we calculated empirical $p$-values by comparing the true correlation value with 10,000 simulated ones obtained by permuting the brain data as shown in Algorithm 1. Given that we are assessing multiple hypotheses simultaneously, we also used the Benjamini-Hochberg False Discovery Rate (FDR) [Benjamini and Hochberg, 1995] to correct for multiple comparisons, at level $\alpha = 0.05$.

**Algorithm 1** Permutation test (for one channel, one time window)

---

**Input:** Brain data $D$, Prediction from base model $P_1$, Prediction from fine-tuned model $P_2$
$D$, $P_1$, and $P_2$ are all of size $(1, N)$, where $N$ is the number of words in the dataset.
**Output:** $pvalue$
$X = \mathrm{corr}(D, P1) - \mathrm{corr}(D, P2)$
$Counter = 0$
**for** $i = 1$ **to** $10,000$ **do**
   $D_i = \mathrm{permute}(D)$
   $X_i = \mathrm{corr}(Di, P1) - \mathrm{corr}(Di, P2)$
   **if** $X_i > X$ **then**
      $Counter = Counter + 1$
   **end if**
**end for**
$pvalue = \frac{Counter+1}{10,000+1}$

---

## K   Comparison between Fine-tuned models and the Base Model

We provide a detailed view of the comparison between the base language model and the models fine-tuned on social (Figure 11) and physical (Figure 12) datasets with each channel plotted.

## L   Additional Control Experiments on MSE Improvement

We conducted additional control experiments to evaluate whether MSE improvement is specific to words that match the category of the dataset on which the model was fine-tuned. Specifically, we evaluated the improvement of physical words on the model fine-tuned on the social dataset, and vice versa.

This analysis reveals that the performance of the model fine-tuned on the social dataset does not significantly differ when assessed with physical and non-physical words (Figure 13B). This finding implies that the enhancements observed are specifically tied to social words. On the other hand, in the model fine-tuned on the physical dataset, we observed a marginal, though not statistically significant, boost in performance with social words (Figure 13C). We propose that this marginal improvement could be attributed to the presence of social and emotional knowledge embedded within the physical dataset. To substantiate this hypothesis, we conducted a thorough review of the physical dataset and identified items that indeed pertain to social or emotional scenarios.

Examples of such items include:

- how do you give a surprise party?
- To help your child feel less afraid when they're going to sleep
- How can you get a child to smile in a photo?
- To help a friend feel better when they are sad
- how to avoid danger?
- To determine if someone has romantic feelings for you

These findings led us to conclude that the marginal improvement in processing social words by the model fine-tuned on the physical dataset may result from exposure to social and emotional content.

## M   Cross-validation on language modelling task

We perform 3-fold cross-validation on the remaining chapters of the *Harry Potter* book (excluding chapters 9 and 10), where we randomly shuffle paragraphs and assign to train:validation:test sets respectively 77%, 16.5%, and 16.5% of the data. Paragraphs that exceeded the context length were excluded. Both the base GPT-2 XL model as well as each model fine-tuned on the three domains were trained to predict the next word for 3 epochs, with the same hyperparameters used in Appendix G.

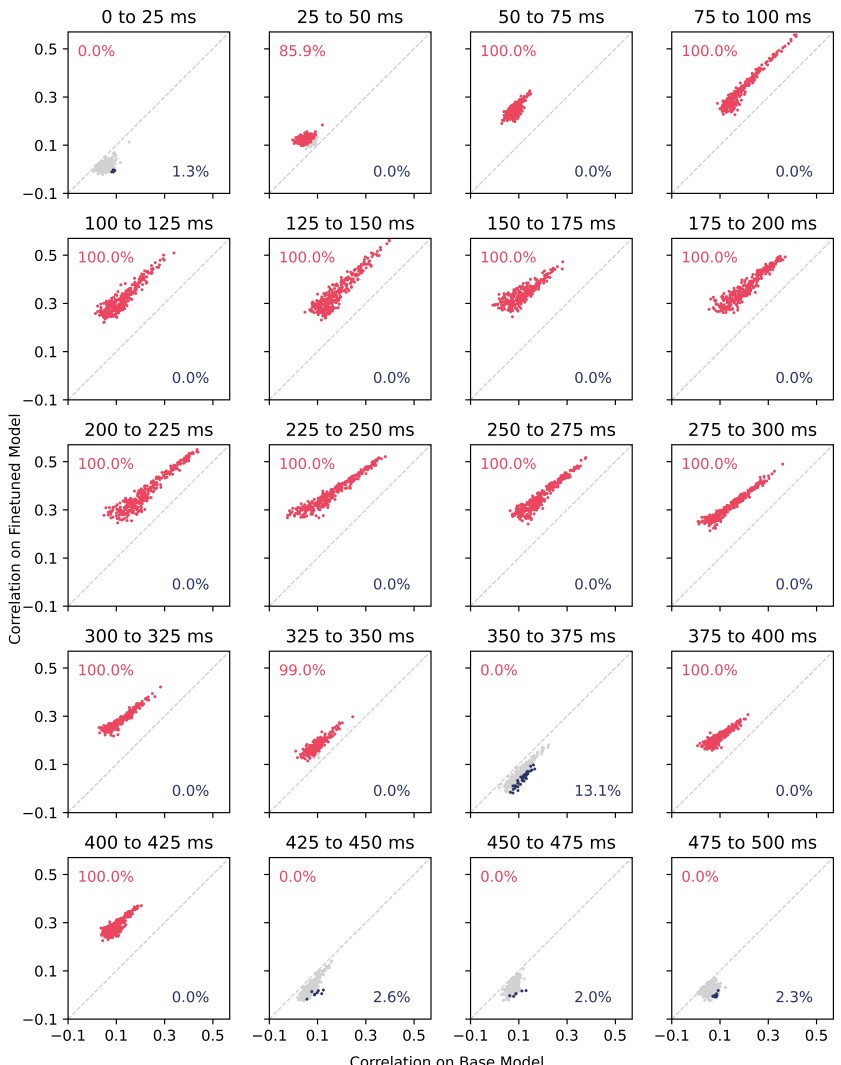

Figure 11: Performance evaluation of the model fine-tuned on the Social IQa (social and emotional) dataset versus the base model using Pearson correlation. Each dot represents a MEG channel. Red channels indicate better predictions by the fine-tuned model, blue channels indicate better predictions by the base model, and gray dots denote non-significant differences. The fine-tuned model outperforms the base model in predicting most channels during language processing time windows.

Results on the test set for each fold are listed below. The average negative-log-likelihood loss per token at the end of training is reported in Table 8.

Table 8: Summary of language-modeling loss across cross-validation folds for models on the remaining chapters of *Harry Potter*.

| Model | Avg. Loss (%) $\pm$ St.dev | Fold 1 Loss | Fold 2 Loss | Fold 3 Loss |
|---|---|---|---|---|
| Base | $0.08795 \pm 0.01707$ | 0.09794 | 0.06391 | 0.1020 |
| Social | $0.1148 \pm 0.00286$ | 0.1119 | 0.1187 | 0.1138 |
| Physical | $0.1001 \pm 0.00184$ | 0.1019 | 0.1009 | 0.0976 |

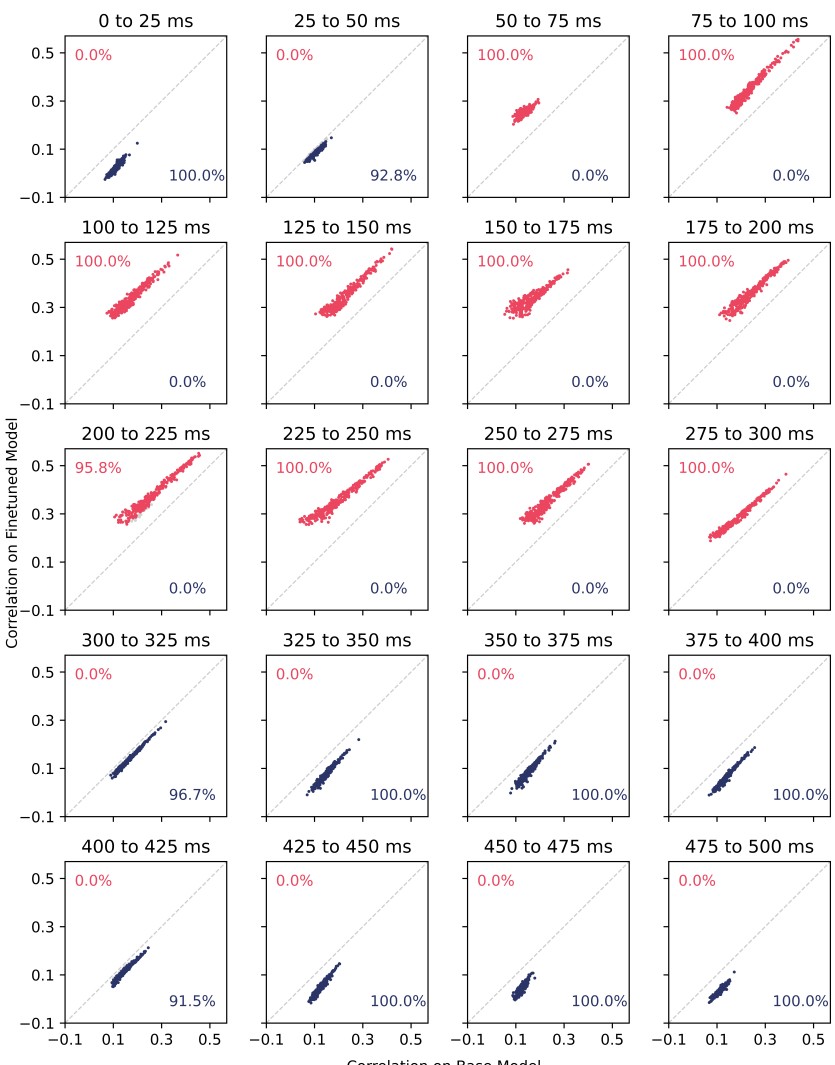

Figure 12: Performance evaluation of the model fine-tuned on the PiQA (physical) dataset versus the base model using Pearson correlation. Each dot represents a MEG channel. Red channels indicate better predictions by the fine-tuned model, blue channels indicate better predictions by the base model, and gray dots denote non-significant differences. The fine-tuned model outperforms the base model in predicting most channels during language processing time windows.

# N    Societal Impacts

This study represents a significant intersection between Neuroscience and Machine Learning, striving to push the boundaries of machine learning models while deepening our understanding of how the human brain functions. In a broader context, this research lays the groundwork for future breakthroughs in the field of neuroscience and for making human-computer interfaces more efficient and intuitive.

However, the development of human-computer interfaces may cause problems in privacy and security, as these interfaces often require the collection and processing of personal data, increasing the risk of data breaches and unauthorized access. There are also ethical concerns regarding the potential for surveillance and the impact on employment, as advanced interfaces might automate tasks currently performed by humans, potentially displacing workers.

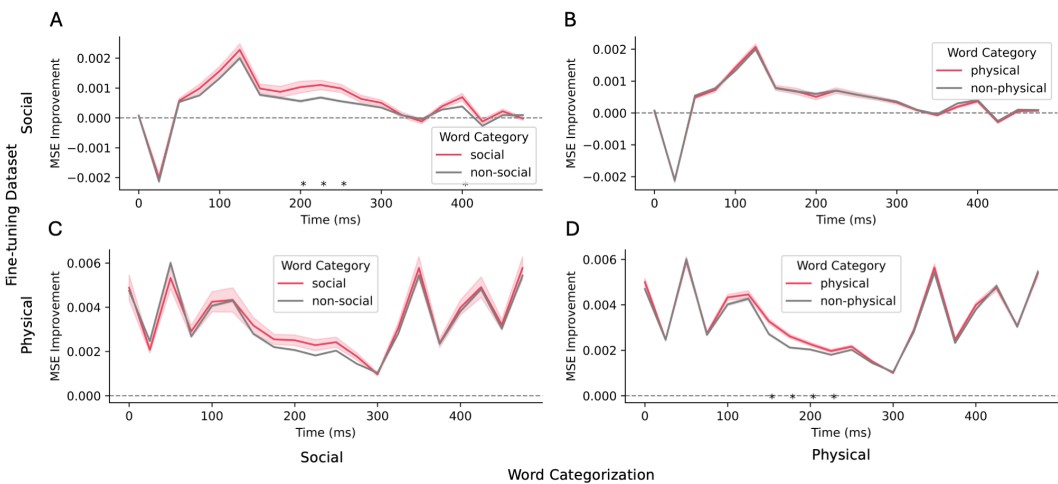

Figure 13: Comparison of improved MSE for A) social vs. non-social words on the social model, B) physical vs. non-physical words on the social model, C) social vs. non-social words on the physical model, and D) physical vs. non-physical words on the physical model. Positive values denote lower MSEs in the fine-tuned model. Shaded region indicates standard error. Asterisks denote time points with significant differences between the two groups (Student's t-test with FDR correction, $p$=0.05).

