# OpenReview forum: "Divergences between Language Models and Human Brains"
_NeurIPS.cc/2024/Conference — NeurIPS 2024 poster_

### Official Review · Reviewer_fRry · 2024-06-20

**Soundness:** 2
**Presentation:** 3
**Contribution:** 3
**Rating:** 7
**Confidence:** 4

**Summary:**

The paper studies the differences in language representations in LLMs (GPT-2 XL) and the human brain (MEG activity), by using LLM representations to predict MEG activity. They use an LLM-based approach to identify two domains where LLM representations do not capture brain activity well: social/emotional intelligence and physical commonsense. They also show that fine-tuning GPT-2 XL on these domains can improve their alignment with human brain responses.

**Strengths:**

From most to least significant:
1. The paper studies the highly interesting, important, and relevant topics of social/emotional intelligence and physical commonsense. Their results highlight the importance of improving social/emotional intelligence and physical commonsense to build more human-brain-aligned LLMs.
2. Interesting methodology using proposer and verifier LLMs. LLMs are growing increasingly capable, and it is great that the paper takes advantage of this trend, to advance the study of the brain.
3. After their analyses on brain alignment, I appreciate that they performed further investigations: human behavioral experiments (Section 3.3), and releasing annotations as a further resource to the dataset (Section 5).

**Weaknesses:**

From most to least significant:
1. They used a relatively low layer (layer 7) of GPT2-XL when making the claim that LMs have poor brain alignment for high-level properties (social/emotional intelligence, physical commonsense). They should have studied a higher layer instead/too.
- The paper argues they identified two domains that LMs do not capture well: social/emotional intelligence and physical commonsense.
- However, they used layer 7 of GPT-2 XL, a relatively low layer since GPT-2 XL has 48 layers. They say they use this because it is the best at predicting brain activity, but this seems much lower than prior research: Layer 8 of 12 in BERT and Layer 12 of 19 in T-XL [1]
- Prior work has shown that lower layers of LLMs capture lower-level linguistic properties, while higher layers capture higher-level properties. Prior work also showed that "LM representations can predict the hierarchy of brain responses" (lines 290-291 in their paper).
- In this case, is it possible that the higher layers (30+) of the same GPT-2 XL model will achieve high brain alignment for social/emotional intelligence and physical commonsense?
- However, re-evaluating more layers is expensive, and I am not requesting them to do this if they can address the comment in writing.
2. Results in Table 1 are not convincing in supporting their overall claims. The LLM-based system of proposers and verifiers generated hypotheses on how the top 100 least-predicted sentences differ from the top 100 well-predicted sentences. However, it seems that 331 of 708 human participants did not agree with the top hypotheses suggested by the LLM system (Table 1), i.e., either found the sentences to be "Equal" or "Convergent". This casts doubt that the two sets of sentences actually differ in terms of social/emotional intelligence and physical commonsense. This, in turn, casts doubt on their overall claim that LLMs do not capture social/emotional intelligence and physical commonsense well.
3. Additional training data as a potential confound
- The paper showed that training LLMs on a dataset for emotional intelligence improves brain alignment. I agree that by itself this is interesting. However, is this because of: (A) statistical similarities between the text in this training dataset and the MEG stimuli, especially for words related to emotional intelligence, or (B) LLM gains better emotional intelligence understanding, as labels were provided and it learns to predict emotions correctly?
- Alternative reasons that finetuning on social dataset improved brain alignment on social words: (a) because the model was provided additional training data? (b) because the additional dataset was higher quality?
- Possible control experiment: training the model (using language modeling) on the same dataset but without associating each question with the correct label. Basically, do not tell the model which is the correct answer (no supervision).
- It is possible that the authors can answer these questions using their further experiments in Appendix I and M. Still, I hope they explicitly address, within the paper, the issue of additional training data as a potential confound.

[1] M. Toneva and L. Wehbe, “Interpreting and improving natural-language processing (in machines) with natural language-processing (in the brain).” 2019

**Questions:**

1. Why is layer 7 of GPT-2 XL the best at predicting brain responses for this dataset? Layer 7 of 48 seems relatively low as opposed to intermediate, whereas prior research showed that intermediate layers are best (lines 115-116 in your paper).

**Limitations:**

The limitations I identified have been mentioned in the paper.

---

> ### Author Rebuttal · Authors · 2024-08-06
>
> Thank you Reviewer fRry for the valuable and constructive feedback! Please find our answers to your comments below.
>
>
> ## Best Performing Layer
>
> Regarding the question about why layer 7 (0-indexed) is the best layer at predicting brain responses in our study, which differs from prior literature, we would like to clarify two main differences between the prior research and our study.
>
> First, the conclusion from [1] is based on fMRI data, whereas our experiments are based on MEG data. Previous study [2] has demonstrated that MEG detects activity primarily in temporal areas, whereas fMRI reveals activity in both temporal and prefrontal regions during language-related tasks. This suggests that the discrepancy in the best-performing layer may arise from the different aspects of brain activity captured by these imaging techniques. However, prior research has consistently shown MEG's effectiveness in detecting the processing of emotions [3-6], supporting its value for studying social and emotional processing.
>
>
> Second, the models used in our study have significantly more parameters than those used in prior research. [1] used models like BERT (110M parameters) and Transformer-XL (128M parameters), while our study employed larger models: GPT-2 XL (1.5B parameters) and Llama-2 7B (7B parameters). One possibility is that as model size increases, the best performing layer tends to be relatively earlier. For instance, in GPT-2 XL, layer 7 is the best (Figure 5A), while in Llama-2 7B, it is layer 3 (Figure 5B).
>
>
> We also want to highlight that we ran our analyses on the **last layer of GPT-2 XL** (Appendix D.1). The generated hypotheses included those observed from the best performing layer, but also encompassed themes such as magical elements and figurative language. This suggests a layer-dependent diversity in encoded information. Nonetheless, it appears that across layers, there is a consistent divergence of physical and social knowledge between the LM and human brains.
>
>
> ## Human Study
>
> Regarding the question about the human study, we would like to clarify the experimental setting. On each trial, participants were presented with one hypothesis and asked to select between one sentence from the divergent set and one sentence from the convergent set. There are a total of ten hypotheses, and each sentence may only satisfy one or two of them. When the hypothesis does not match the sentence from the divergent set, participants should have no preference for either sentence. Thus, the absolute values are not meaningful; the key point is the comparison of the percentage of responses preferrring "divergent" between the top and bottom hypotheses conditions, represented by the blue areas in Figure 10A and Figure 10B. Thank you for bringing this to our attention, we'll will ensure that we clarify these points about the study in the reversion of the paper.
>
> ## Fine-Tuning
>
> Regarding the question about additional training data as a potential confound in fine-tuning, we appreciate your insights. To address this, we conducted two control checks. First, we confirmed that the model's improved brain alignment is not due to increased language modeling ability (Appendix M). Second, we verified the domain specificity of fine-tuning by evaluating the performance of physical words in the model fine-tuned on the social dataset (Appendix L). Our experiments show that the model's performance on physical versus non-physical words does not differ significantly, indicating that the improvement is specific to social knowledge (Figure 13B). Similar results were found when evaluating the model fine-tuned on the physical dataset with social versus non-social words (Figure 13C). We appreciate your raising these points and will explicitly address the potential confound of additional training data in the revised version of the paper.
>
>
>
> [**1**] Toneva, Mariya, and Leila Wehbe. "Interpreting and improving natural-language processing (in machines) with natural language-processing (in the brain)." Advances in neural information processing systems 32 (2019).
>
> [**2**] Billingsley-Marshall, Rebecca L., et al. "A comparison of functional MRI and magnetoencephalography for receptive language mapping." Journal of Neuroscience Methods 161.2 (2007): 306-313.
>
> [**3**] Giorgetta, Cinzia, et al. "Waves of regret: A meg study of emotion and decision-making." Neuropsychologia 51.1 (2013): 38-51.
>
> [**4**] Peyk, Peter, et al. "Emotion processing in the visual brain: a MEG analysis." Brain topography 20 (2008): 205-215.
>
> [**5**] Dumas, Thibaud, et al. "MEG evidence for dynamic amygdala modulations by gaze and facial emotions." PloS one 8.9 (2013): e74145.
>
> [**6**] Hagan, Cindy C., et al. "MEG demonstrates a supra-additive response to facial and vocal emotion in the right superior temporal sulcus." Proceedings of the National Academy of Sciences 106.47 (2009): 20010-20015.

---

> > ### Comment · Reviewer_fRry · 2024-08-08
> >
> > I will raise my score from 6 to 7. The authors provided convincing clarifications for most of the weaknesses (and their sub-points) I raised.
> > 1. Best Performing Layer (resolved)
> > - Thanks for the clarification on MEG vs fMRI, and larger vs smaller models.
> > 2. Human Study (resolved)
> > - Thanks for the clarification, especially that "There are a total of ten hypotheses, and each sentence may only satisfy one or two of them." This helped me to understand why so many participant ratings are "Equal" or "Convergent", even for the set of top hypotheses.
> > 3. Fine-Tuning (resolved)
> > - Thanks for the clarifications.

---

### Official Review · Reviewer_b2dQ · 2024-07-12

**Soundness:** 4
**Presentation:** 4
**Contribution:** 3
**Rating:** 7
**Confidence:** 4

**Summary:**

Language models are known to predict MEG signals in humans during reading. In this work, the authors explored for what "kinds" of texts are language models bad at predicting MEG signals. The authors used a novel method to propose possible hypotheses, and found multiple possible categorizations of weak prediction texts. Focusing on social knowledge and spatial commonsense, the authors find that finetuning the language model on texts from these respective domains improves the prediction accuracy.

**Strengths:**

1. The experiment design is clean and straightforward.
2. The detailed evaluation of error pattern during brain signal prediction is very valuable.
3. The automated method of hypothesis proposal might be generalizable to other forms of experiments in cognitive sciences as well.

**Weaknesses:**

Many aspects mentioned in the work where the language model are not performant on, for example common sense and social reasoning, have large leaps forward in more modern models. A future work could focus on using more modern language models with stronger capacity.

**Questions:**

1. Temporal signal prediction is convincing, another aspect that is interesting is the spectral structure of the predictions.
2. Do layers other than 7 display the same pattern of lower prediction accuracy? Do you observe this consistently across layers, or just at layer 7?

**Limitations:**

Properly addressed.

---

> ### Author Rebuttal · Authors · 2024-08-06
>
> Thank you Reviewer b2dQ for the positive and constructive review! Please find our answers to your comments below.
>
> >**Many aspects mentioned in the work where the language model are not performant on, for example common sense and social reasoning, have large leaps forward in more modern models. A future work could focus on using more modern language models with stronger capacity.**
>
>
> Thank you for your comment, we agree that more modern models may demonstrate enhanced capabilities in areas like common sense and social reasoning. To explore this, we replicated our analyses on Llama-2 7B, a larger and more recent model compared to GPT-2 XL (see Appendix E). Our results show that Llama-2 7B’s hypotheses predominantly focus on physical knowledge, with the social and emotional dimension no longer a theme in the generated hypotheses. This suggests that Llama-2 7B might possess a more sophisticated understanding of social and emotional contexts. We look forward to replicating the analyses on other models with different parameter counts, pre-training datasets, and methodologies for training and fine-tuning in future work.
>
> >**Temporal signal prediction is convincing, another aspect that is interesting is the spectral structure of the predictions.**
>
>
> Yes, we think exploring spectral dimensions would be an intriguing direction for future research. For example, alpha waves are often associated with passive attention, whereas higher frequency waves, such as gamma waves, are linked to concentration and information processing. We’re excited about the possibility of examining how brain signals from different frequency bands might correlate with language model embeddings.
>
> > **Do layers other than 7 display the same pattern of lower prediction accuracy? Do you observe this consistently across layers, or just at layer 7?**
>
>
> We also ran the analyses on the last layer of GPT-2 XL (Appendix D.1). The generated hypotheses included those observed from the best performing layer, but also encompassed themes such as magical elements and figurative language. This suggests a layer-dependent diversity in encoded information. Nonetheless, it appears that across layers, there is a consistent divergence of physical and social knowledge between the LM and human brains.

---

> > ### Comment · Reviewer_b2dQ · 2024-08-09
> >
> > My questions are adequately addressed. I will maintain my current score.

---

### Official Review · Reviewer_jRnr · 2024-07-14

**Soundness:** 3
**Presentation:** 3
**Contribution:** 2
**Rating:** 6
**Confidence:** 4

**Summary:**

This paper explores the differences between LMs and the human brain in processing language. The authors conduct experiments using MEG data from reading and listening tasks to investigate elements of MEG responses that LMs cannot adequately explain. LLMs are used to automatically generate hypotheses, identifying the domains where LMs lack knowledge compared to the human brain. The authors then fine-tune LMs on these specific domains to improve the LMs-brain alignment.

**Strengths:**

The idea of assessing and interpreting the biological plausibility of LMs is a feasible way to enhance both their interpretability and model design. Focusing on the divergences between LMs and the brain is currently lacking in the field.

The paper is well-written, includes good visualizations, and the idea is easy to follow with good reproducibility.

**Weaknesses:**

Only two LMs are used in the experiment. Additionally, GPT-2 is somewhat outdated compared to recently released open-source LLMs, as it shows limited language understanding and reasoning capability, making it far from achieving human-level AGI. The focus should be more on LLMs such as Mixtral and Gema.

More details about the prompts and API usage should be discussed for the proposer and verifier LLMs.

**Questions:**

The relatively narrow datasets used in the paper, focusing on human dialogue and stories, which intuitively consist of emotion representation, could easily introduce bias.

**Limitations:**

See Weakness and Questions

---

> ### Author Rebuttal · Authors · 2024-08-06
>
> Thank you Reviewer jRnr for the valuable and constructive feedback! Please find our answers to your comments below.
>
> ## GPT-2 vs Modern LLMs
>
> We agree that more modern models may demonstrate enhanced capabilities in areas like language understanding and social reasoning. To explore this, we replicated our analyses on Llama-2 7B, a larger and more recent model compared to GPT-2 XL (see Appendix E). Our results show that Llama-2 7B’s hypotheses predominantly focus on physical knowledge, with the social and emotional dimension no longer a theme in the generated hypotheses. This suggests that Llama-2 7B might possess a more sophisticated understanding of social and emotional contexts. We look forward to replicating the analyses on other models with different parameter counts, pre-training datasets, and methodologies for training and fine-tuning in future work.
>
> ## Proposer and Verifier
>
> Thanks for bringing up the importance of discussing more details about prompts and API usage for the proposer and verifier LLMs. We have included the prompt we used and will incorporate these details into the next version of the paper.
>
> We used the gpt-4-turbo-instruct as the proposer model. The prompt for the proposer is:
>
> > {A_block}
> >
> > {B_block}
> >
> > The dataset includes two chapters from "Harry Potter and the Sorcerer's Stone". The two groups are generated based on the difference between language model and human responses to these sentences. The Group A snippets sentences where language models and humans show divergent responses, while the Group B snippets sentences where language models and humans show similar responses.
> >
> > I am a literary analyst investigating the characteristics of words. My goal is to figure out which sentences induce different responses for language models and human responses.
> >
> > Please write a list of hypotheses about the datapoints from Group A (listed by bullet points "-"). Each hypothesis should be formatted as a sentence fragment. Here are three examples.
> >
> > \- "{example_hypothesis_1}"
> >
> > \- "{example_hypothesis_2}"
> >
> > \- "{example_hypothesis_3}"
> >
> > Based on the two sentence groups (A and B) from the above, more sentences in Group A ...
>
>
> We used FLAN-T5-XXL as the validator. The prompt for the validator is:
>
> > Check whether the TEXT satisfies a PROPERTY. Respond with Yes or No. When uncertain, output No.
> >
> > Now complete the following example -
> >
> > input: PROPERTY: {hypothesis}
> >
> > TEXT: {text}
> >
> > output:
>
>
> ## Datasets
>
> Thank you for pointing out the concern about the relatively narrow datasets used in the paper. While we did discuss it in the final paragraph of the Conclusions, Limitations, and Future Work section, it is a limitation that we plan to address in future work. Due to the high cost of recording human brain responses with neural imaging techniques, there is a very limited selection of publicly available datasets. We agree that incorporating datasets from a broader range of contexts in future studies would be valuable for further validating and expanding our findings. Indeed, to our knowledge, several new relevant datasets are currently in progress, and we also plan to collect our own data using more diverse texts.

---

> > ### Comment · Reviewer_jRnr · 2024-08-12
> >
> > I have read the authors' rebuttal which has addressed my concerns to some degree. I have raised my rating accordingly. Thanks.

---

### Official Review · Reviewer_hoiu · 2024-07-15

**Soundness:** 3
**Presentation:** 3
**Contribution:** 3
**Rating:** 7
**Confidence:** 4

**Summary:**

The authors use a data-driven method to generate hypotheses about particular words/linguistic contexts in which a brain encoding model does not accurately predict the brain response to language. they find that social/emotional/physical complexity, along with linguistic complexity, are all associated with worse brain encoding performance. They fine-tune the base GPT model to perform better at social/emotional reasoning tasks and find that this yields an improvement in brain encoding for these particular words within the relevant language processing window (~75-400 ms).


Update: I have read the other reviews and author rebuttals and will keep my score as-is.

**Strengths:**

- This is an interesting and novel approach using LLM proposer-verifier along with a behavioral experiment to validate the labeling generated by models.
- The thorough experimental pipeline shows that a fine-tuning intervention designed along these hypotheses improves brain encoding performance as expected.
- The paper opens up a new mysterious result to be explored -- it looks like there's a tradeoff between fitting these temporally intermediate responses and very early and very late responses to words.

**Weaknesses:**

- The fine-tuning intervention needs to be appropriately baselined, for example by fine-tuning on other tasks which don't match the hypotheses about what is driving brain encoding performance. If fine-tuning on an appropriate counterfactual sample doesn't improve brain encoding performance, this will strengthen the positive finding. (I'm not totally clear on the discussion in Appendix M -- are you saying that fine-tuning through language modeling on Harry Potter doesn't yield brain encoding improvements? This is a step in the right direction, I think.)
- The effect sizes of improvements here are very small; the results would be much stronger if you could reproduce this effect on a second dataset.

**Questions:**

It seems like there is a lot more going on in the example sentences (Appendix C) than the hypotheses cover. For example, many function words are colored as "most divergent" despite not fitting the leading hypotheses. Can you provide a broader sample that might convince the reader of the coverage of the model-derived hypotheses?

**Limitations:**

Yes

---

> ### Author Rebuttal · Authors · 2024-08-06
>
> Thank you Reviewer hoiu for the positive and valuable comments! Please find our answers to your comments below.
>
> ## Fine-tuning
>
> > **The fine-tuning intervention needs to be appropriately baselined, for example by fine-tuning on other tasks which don't match the hypotheses about what is driving brain encoding performance. If fine-tuning on an appropriate counterfactual sample doesn't improve brain encoding performance, this will strengthen the positive finding.**
>
> Thank you for highlighting the need to baseline the fine-tuning results on unrelated tasks. While we didn't conduct the exact experiment you suggested, we did run two related experiments as control checks. First, we confirmed that the model's improved brain alignment is not due to increased language modeling ability (Appendix M). Second, we verified the domain specificity of fine-tuning by evaluating the performance of physical words in the model fine-tuned on the social dataset (Appendix L). Our experiments show that the model's performance on physical versus non-physical words does not differ significantly, indicating that the improvement is specific to social knowledge (Figure 13B). Similar results were found when evaluating the model fine-tuned on the physical dataset with social versus non-social words (Figure 13C).
>
> >**The effect sizes of improvements here are very small; the results would be much stronger if you could reproduce this effect on a second dataset.**
>
> We appreciate your suggestion to replicate the fine-tuning results on a different dataset. However, due to the limited rebuttal period, we are unable to conduct this experiment as it requires recruiting human annotators to annotate each word in the second dataset. We greatly appreciate your input and look forward to incorporating datasets from a broader range of contexts in future studies to further validate and expand our findings.
>
>
> ## Example Sentences
>
> > **It seems like there is a lot more going on in the example sentences (Appendix C) than the hypotheses cover. For example, many function words are colored as "most divergent" despite not fitting the leading hypotheses. Can you provide a broader sample that might convince the reader of the coverage of the model-derived hypotheses?**
>
> Please refer to the PDF in the global rebuttal for 10 additional sentences selected from the top 20% most divergent sentences. We note that identifying hypotheses by examining individual words in sentences is challenging, which motivates our use of an automatic hypothesis proposing method. However, it is worth noting that words related to emotions (e.g., pleased, nervous, afraid, madly) are often among the most divergent.

---

### Author Rebuttal · Authors · 2024-08-06

Please refer to the attached PDF for 10 additional sentences colored based on prediction error. These sentences are selected from the top 20% most divergent sentences in the Harry Potter dataset. Each of the five colors corresponds to a 20-percentile range of words from the entire dataset.

---

### Decision · Program_Chairs · 2024-09-25

**Decision:**

Accept (poster)

**Comment:**

The authors employ a data-driven approach to identify specific words and linguistic contexts where a brain encoding model fails to accurately predict neural responses to language. They discover that factors such as social, emotional, physical, and linguistic complexity contribute to poorer performance of the brain encoding model. To address this, they fine-tune the base GPT model to enhance its capabilities in social and emotional reasoning tasks. This adjustment leads to improved brain encoding predictions for these specific words during the relevant language processing time frame, approximately 75 to 400 milliseconds.

The reviewers raised many technical issues, including experiments with only two models, the possibility that newer models would demonstrate different performance and various technical questions about the experiments. Some of these issues have been solved in the discussion period and all four reviewers are leaning towards acceptance. Overall, this is an empirical paper on an important topic and I believe it makes a nice contribution that justifies publication.

As a side note - the paper is missing a very relevant citation that also shows that fine tuning to a specific domain can substantially improve alignment with the human brain, although with fMRI data:

Tikochinski, R., Goldstein, A., Yeshurun, Y., Hasson, U., & Reichart, R. (2023). Perspective changes in human listeners are aligned with the contextual transformation of the word embedding space. Cerebral Cortex, 33(12), 7830-7842.‏

I strongly encourage the authors to cite and compare to this paper.